

# Prognostic value of a modified pathological staging system for gastric cancer based on the number of retrieved lymph nodes and metastatic lymph node ratio

Guiru Jia, Dagui Zhou, Xiao Tang, Jianpei Liu and Purun Lei

Department of Gastrointestinal Surgery, Third Affiliated Hospital of Sun Yat-Sen University, Guangzhou, Guangdong, China

## ABSTRACT

**Aim:** The prognosis for gastric cancer (GC) remains grim, underscoring the importance of accurate staging and treatment. Given the potential benefits of using lymph node ratio (LNR) for improved prognostication and treatment planning, it is critical to incorporate examined lymph nodes (ELN) count in an integrated GC staging system.

**Methods:** Patients data from the Surveillance, Epidemiology, and End Results (SEER) database between 2010 and 2015 was utilized as training set. The Mantel-Cox survival test was used to calculate chi-square values for 40 LNR segments with a 0.025 interval, defining a novel LNR-based N (rN) classification based on the cutoff points. A revised AJCC (rAJCC) staging system was established by replacing the 8th AJCC N staging with a rN classification. The relationship between the ELN count and prognosis or positive lymph node detection was conducted by using multivariable models. The series of the odds ratios and hazard ratios were fitted with a locally weighted scatterplot smoothing (LOWESS) smoother, and the structural break points were determined by Chow test to clarify an optimal minimum ELN count. The integrated GC staging system incorporated both rAJCC system and the ideal ELN count. Discriminatory ability and prognostic homogeneity of the rAJCC and integrated staging system was compared with AJCC staging system in the SEER validation set (2016–2017), the Cancer Genome Atlas Program (TCGA) database, and the Third Affiliated Hospital of Sun Yat-sen University database.

**Results:** The current study found that LNR and ELN count are both significantly associated with the prognosis of GC patients (HR = 0.98, $p < 0.001$ and HR = 2.51, $p < 0.001$). Four peaks of the chi-square value were identified as LNR cut-off points at 0.025, 0.175, 0.45 and 0.6 to define a novel rN stage. In comparison to the 8th AJCC staging system, the rAJCC staging system demonstrated significant prognostic advantages and discriminatory ability in the training set (5-Y OS AUC: 71.7 *vs.* 73.0; AIC: 57,290.7 *vs.* 57,054.9). The superiority of the rAJCC staging system was confirmed in all validation sets. Using a LOWESS smoother and Chow test, a threshold ELN count of 30 was determined to maximum improvement in the prognosis of node-negative patients without downgrading due to potential metastasis, while also maximizing the detection efficiency of at least one involved lymph node. The integrated staging system, combining the refined rAJCC classification with an optimized ELN count threshold, has demonstrated superior

Corresponding authors
Jianpei Liu, kamplau@126.com
Purun Lei, leipurun@163.com

discriminatory performance compared to the standalone rAJCC or the traditional AJCC system.

**Conclusion:** The development of a novel GC staging system, which integrated the LNR-based N classification and the minimum ELN count, has exhibited superior prognostic accuracy, holding promise as a valuable asset in the clinical management of GC. However, it is crucial to recognize the limitations from the retrospective database, which should be addressed in subsequent analyses.

## INTRODUCTION

Gastric cancer (GC) is the fifth most common cancer globally (*Bray et al., 2024*) and the sixth most common malignancy in China (*Zheng et al., 2024*). Radical resection, in conjunction with adjuvant treatment based on pathological staging, provides a potential for a cure, especially in cases of early-stage gastric cancer (*Smyth et al., 2020*; *Rosa et al., 2022*). However, the prognosis remains poor, highlighting the need for accurate staging and appropriate treatment (*Degiuli et al., 2021*).

The tumor-node-metastasis (TNM) staging system by the American Joint Committee on Cancer (AJCC) for GC globally evaluates primary tumor invasiveness and size (T), regional lymph node involvement (N), and absence of distant metastasis (M) (*In et al., 2018*). Lymph node involvement is a significant factor in predicting patients' recurrence and survival, but assessing it solely based on the number of positive lymph nodes can lead to inaccuracy due to incomplete lymphadenectomy (*Kinami, Saito & Takamura, 2022*; *Zeng et al., 2023b*). The lymph node ratio (LNR) denotes the proportion of metastatic regional lymph nodes (LN) to the total number of examined lymph nodes (ELN) obtained from the specimen (*Yamashita et al., 2016*). LNR has been shown to be a better prognostic indicator than the number of positive lymph nodes or the total number of ELNs in multiple malignancies, including gastric cancer (*Kano et al., 2020*; *Kotecha et al., 2022*; *Ergenç et al., 2023*). The utilization of an LNR-based modified staging system has demonstrated a higher accuracy in predicting survival compared to the 8th edition of the AJCC staging system (*Huang et al., 2020*; *Yin et al., 2024*). However, all previous studies have overlooked the significance of total ELN in accurately staging cancer (*Gu et al., 2020*; *Zeng et al., 2023b*). While AJCC guidelines suggest assessing at least 16 LNs per patient, it is uncertain how many ELN are needed for reliable stage assignment and strong prognostic value (*In et al., 2018*). A higher ELN count can indicate a more thorough lymphadenectomy and aid in detecting metastatic LNs (*Macalindong et al., 2018*). Therefore, a combined assessment of both ELN count and lymph node involvement evaluation is required.

The Surveillance, Epidemiology, and End Results (SEER) Program collects cancer information that encompass 50% of the U.S. population, including patients with gastric cancer (*Daly & Paquette, 2019*). However, there were no studies that incorporated both ELN and LNR in the revised staging system for gastric cancer. In the current study, the

SEER database was used for establishing a staging system by replacing the 8th AJCC N classification with the LNR classification and incorporating minimum ELN count. Data from the Cancer Genome Atlas Program (TCGA) and Third Affiliated Hospital of Sun Yat-sen University was used for external validation for the novel system.

## METHODS

### Study population and data collection

Clinical data from the US Surveillance, Epidemiology, and End Results (SEER) Program from 2010–2015 (https://seer.cancer.gov/) was extracted and analyzed as training set, data from 2016–2017 was adopted as internal validation set. Data from The Cancer Genome Atlas Program (TCGA) (https://portal.gdc.cancer.gov/) and prognosis data from the gastrointestinal surgery department, Third Affiliated Hospital of Sun Yat-sen University were applied as external validation sets.

Screening criteria for gastric cancer cases were as follow: exclusion of cases with only autopsy or death certificate, cases where initial tumor location was not stomach, patients with stage 0 and stage IV, cases without radical surgery, non-adenocarcinoma cases, death cases within 1 month after operation, and cases with unknown lymph node information and AJCC TNM stage.

The study analyzed various factors such as age of diagnosis (<50 years, 50–69 years, >69 years), gender, race (white, black, other), AJCC T stage (T1–T4b), AJCC TNM stage (I–III), primary tumor location (stomach body, antrum/pylorus, cardia/fundus, greater gastric recurve, lesser gastric recurve, overlapping area, NOS), clinical features such as tumor size (≥5 cm, <5 cm, unknown), tumor grade (I–IV), chemotherapy, radiotherapy, number of lymph nodes retrieved and number of metastases, and lymph node positive rate. The populations of American Indian/Alaskan and Asian/Pacific Islander were classified as "other" due to small sample sizes. Tumor grade was also analyzed, with grades I–IV representing highly differentiated, moderately differentiated, poorly differentiated, and signed-ring cell carcinoma, respectively. Overall survival (OS) is the time from cancer diagnosis to death from any cause, while disease-specific survival (DSS) is the time from cancer diagnosis to death specifically due to the disease.

### Statistical analysis

The chi-square test was used to compare differences between categorical variables, while the t-test was used for continuous variables. Univariate and multivariate Cox regression were used to examine the association between prognosis and the covariates. The Kaplan-Meier method is used to compare overall OS and DSS among groups. Hazard ratios (HR) for mortality were reported following adjusting for covariates including age, year of diagnosis, gender, race, tumor features (site, size, and grade).

The study used Mantel-Cox survival test to calculate chi-square values for 40 segments of LNR with a 0.025 interval, identified four peaks as cutoff points, and defined LNR-based N (rN) classification based on these cutoff points. Bootstraps with 1,000 resamples was conducted to internally validate the rN staging system. Patients included in the study were then redistributed according to the revised AJCC (rAJCC) staging system. We compared

the performance of the rAJCC system with that of the 8th AJCC staging system in terms of discriminatory ability and the prognostic homogeneity. These comparisons were assessed using the area under the receiver operating characteristic curve (AUC), Akaike's information criterion (AIC), and Bayesian Information Criterion (BIC). A lower AIC value or a higher AUC indicates a stronger discriminatory capacity of the staging system.

The proper threshold of ELN count analysis was conducted in two steps. COX regression firstly performed based on OS and DSS to examine the HR of each ELN count in patients with negative LNs. Patients with only one ELN were used as a reference, HR value for each ELN count group for both OS and DSS was analyzed using locally weighted scatterplot smoothing (LOWESS) scatter curve fitting. The minimum count of ELNs was determined by Chow test at which the slope of the curve changes significantly. Logistic regression analysis was then performed based on patients' data with negative or only one LN metastasis, using node status as the outcome variable. Patients with only one ELN was defined as reference. LOWESS and Chow test help draw the odds ratio curve of each ELN count group and defined the ideal cutoff LN count for maximum detection efficacy of at least one involved LN.

The minimum ELN count and the rAJCC staging system merged to develop the integrated GC staging system. We compared the performance of the integrated GC staging system to that of the 8th AJCC staging system in terms of discriminatory ability and prognostic homogeneity. These aspects were evaluated using metrics such as AUC, AIC and BIC.

A significance level of $p < 0.05$ was used for all data analysis. Statistical analyses were conducted using IBM SPSS Statistics for Windows v. 20.0 (IBM Corp., Armonk, NY, USA) and R version 3.6.2 software (Bell Laboratories, Murray Hill, NJ, USA).

The current study is performed following the principles from the declaration of Helsinki. Patient from our center was informed before database enrollment individually, data from the TCGA database were publicly available and de-identified. The study was approval by the institutional review board of the Third Affiliated Hospital of Sun Yat-sen University for human data analysis as No. II2023-062-01.

## RESULTS

### Patient characteristics

The study analyzed data from 8,137 patients in the SEER database, 277 patients in the TCGA database, and 719 patients from the authors' center. The SEER patients were divided into training (2010–2015) and validation (2016–2017) sets based on years of diagnosis. The selection process is shown in Fig. 1. The datasets utilized in our analysis were designated as follows: the training set as dataset 1, the validation set as dataset 2, the TCGA set as dataset 3, and the data from the author's institution as dataset 4.

The majority of patients in the training set, validation set, TCGA set, and data from author's center were over 50 years old (90.35%, 90.15%, 92.42% and 79.28%) and had moderate to poor differentiation (85.72%, 84.15%, 97.12% and 90.41%). In the datasets derived from the SEER database, namely dataset 1 and dataset 2, the majority of patients identified as White, comprising 65.67% and 62.88%, respectively. This demographic

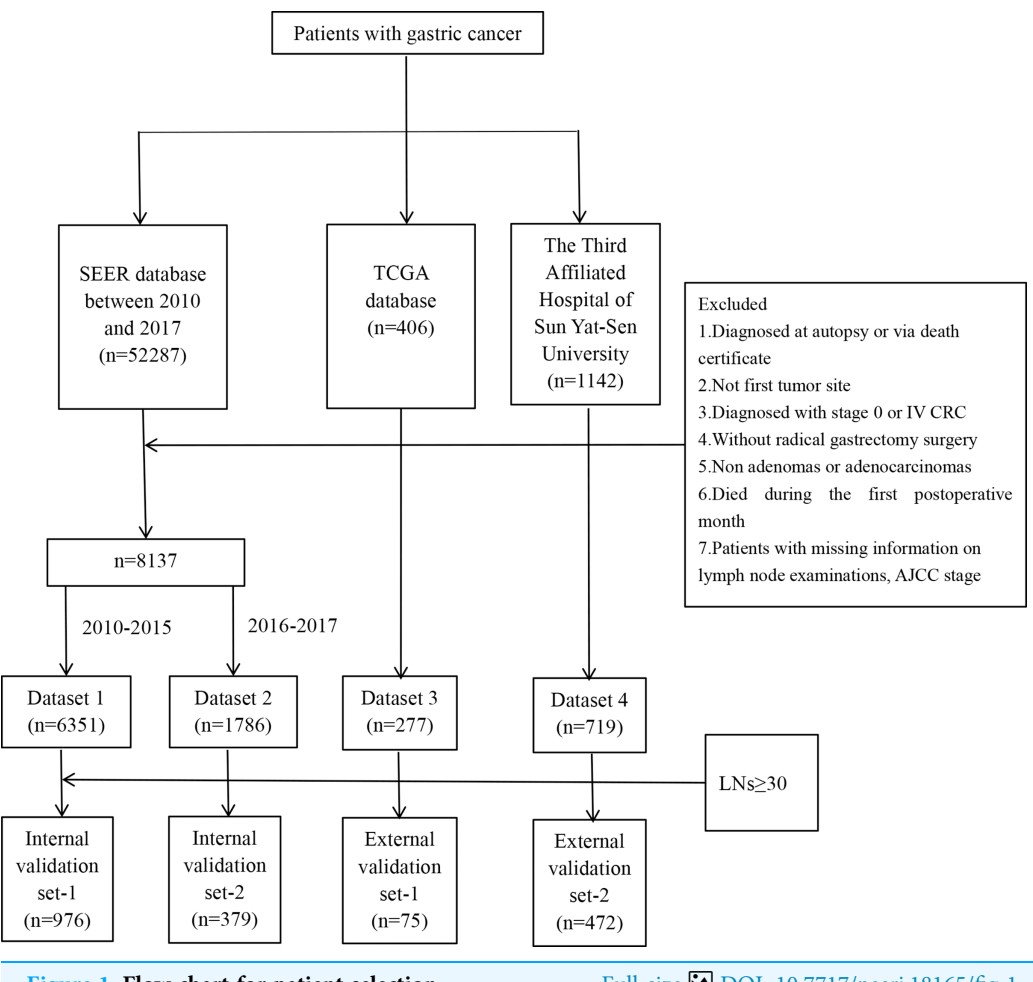

**Figure 1 Flow chart for patient selection.**

distribution was mirrored in the TCGA dataset 3, where the White ethnicity constituted 68.23% of the patient population. In contrast, all patients from our single-center study were of Chinese ethnicity, with no representation from other ethnic groups.

Regarding gastric tumor size, approximately half of the patients in the SEER and TCGA datasets had tumors measuring less than 5 cm, with percentages of 56.54%, 56.66%, and 48.38%, respectively. In our single-center study, this proportion was higher, at 63.00%. The primary tumor sites included the body, antrum/pylorus, cardia/fundus, greater curvature, lesser curvature, overlapping regions, and stomach without detailed specification. Notably, the cardia/fundus was the most common primary tumor site in the SEER and TCGA datasets, whereas in our single-center study, the antrum/pylorus was the predominant site. Most patients were diagnosed with T3 or T4 gastric cancer, accounting for 59.49%, 61.09%, 69.32%, and 67.32% of cases. Furthermore, patients found to have node-positive disease according to the 8th AJCC staging criteria, account for 56.53%, 57.39%, 69.31%, and 60.08% in each group. The median ELN counts were higher in the data from our center (35 [IQR26-45]) than in the other sets (16 [IQR10-24], 18 [IQR12-27], 17[IQR10-31]), while the PLN counts were similar across all sets (1 [IQR 0, 4], 0 [IQR 0, 3], 2 [IQR 0, 7] and 2

[IQR 0, 7]). In terms of treatment modalities, across dataset 1, dataset 2, and dataset 3, 56.95%, 61.42%, 35.02% and 59.25% of patients, respectively, received postoperative chemotherapy. Radiotherapy was administered to 37.77%, 33.26%, and 10.83% of patients in the respective datasets, none underwent radiotherapy in our single center. The median follow-up time was longest in the training set (50 months, [19, 77]) and data from author's center (45 months, [20, 73]). Detailed clinicopathologic characteristics can be found in Table S1.

## Univariate and multivariate analysis

Prognosis data and related variables from the training set were analyzed. As shown in Table 1, univariate analysis revealed that male patients (1.13 [1.05–1.21], $p = 0.001$), age over 69 years old (1.54, [1.36–1.75], $p < 0.001$), non-white/black patients ($p < 0.001$), degree of differentiation ($p < 0.001$), tumor size ≥5 cm (1.68, [1.57–1.80], $p < 0.001$), tumor locate at cardia/fundus (1.26, [1.11–1.44], $p < 0.001$), depth of tumor invasion ($p < 0.001$), N stage ($p < 0.001$), AJCC pathological classification ($p < 0.001$), chemotherapy (1.28, [1.20–1.37], $p < 0.001$), radiotherapy (1.28, [1.19–1.37], $p < 0.001$), ELN count (0.99, [0.99–1.00], $p < 0.001$), PLN count (1.06, [1.06–1.06], $p < 0.001$), and LNR (7.84, [7.05–8.72], $p < 0.001$) were significantly associated with overall survival of patients diagnosed with gastric cancer patients in dataset 1.

The following factors were independently correlated with poorer OS using multivariate analysis, male patients (HR = 1.14, $p = 0.001$); patients between 50 to 69 years old (HR = 1.18, $p = 0.009$) and over 69 years old (HR = 1.88, $p < 0.001$); non-white/black patients (HR = 0.78, $p < 0.001$); tumor diameter ≥5 cm (HR = 1.10, $p = 0.014$); tumors locate at cardia/fundus area (HR = 1.39, $p < 0.001$); with advancing of pT and pTNM classification ($p < 0.05$). Furthermore, chemotherapy could help improve the overall prognosis (HR = 0.76, $p < 0.001$). Higher ELN count was significantly associated with better prognosis of GC (HR = 0.98, $p < 0.001$), while increasing PLN count and LNR were adverse prognostic factors (HR = 1.03, $p < 0.001$ and HR = 2.51, $p < 0.001$). The detailed data was shown in Table 1.

## LNR classification and revised GC staging evaluation

LNR was then divided into 40 segments using a 0.025 interval, and chi-square values were calculated through Mantel-Cox survival test between adjacent segments in patients from the training set. Four peaks of the chi-square value were identified as cut-off points at 0.025, 0.175, 0.45, and 0.6 ($p < 0.05$), as shown in Table S2. The rN classification was used to distinguish five groups: rN0 ($0 \leq$ LNR $< 0.025$), rN1 ($0.025 \leq$ LNR $< 0.175$), rN2 ($0.175 \leq$ LNR $< 0.45$), rN3a ($0.45 \leq$ LNR $< 0.6$), and rN3b ($0.6 \leq$ LNR $\leq 1$) based on LNR values. An 8th AJCC N classification was replaced with the corresponding rN classification to develop a rAJCC staging system with rI, rII, and rIII stages.

According to the 8th AJCC, N classification was divided into N0, N1, N2, N3a, and N3b with percentages of 2,761 (43.47%), 1,558 (24.53%), 1,044 (16.44%), 717 (11.29%) and 271 (4.27%), respectively. In the revised classification system, the rN categories were

Table 1 Univariate and multivariate Cox analyses of overall survival (OS) in dataset 1.

| (OS) Characteristics | Univariate analysis | | Multivariate analysis | |
|---|---|---|---|---|
| | HR (95% CI) | *p*-value | HR (95% CI) | *p*-value |
| Sex | | | | |
| Female | 1 | Ref. | 1 | Ref. |
| Male | 1.13 [1.05–1.21] | 0.001 | 1.14 [1.06–1.23] | 0.001 |
| Age | | | | |
| <50 | 1 | Ref. | 1 | Ref. |
| 50–69 | 1.09 [0.96–1.23] | 0.190 | 1.18 [1.04–1.34] | 0.009 |
| >69 | 1.54 [1.36–1.75] | <0.001 | 1.88 [1.66–2.14] | <0.001 |
| Year | | | | |
| 2010 | 1 | Ref. | / | / |
| 2011 | 1.09 [0.97–1.21] | 0.137 | / | / |
| 2012 | 0.99 [0.89–1.11] | 0.882 | / | / |
| 2013 | 1.01 [0.91–1.14] | 0.799 | / | / |
| 2014 | 0.99 [0.89–1.12] | 0.927 | / | / |
| 2015 | 1.04 [0.92–1.18] | 0.506 | / | / |
| Race | | | | |
| White | 1 | Ref. | 1 | Ref. |
| Black | 0.99 [0.90–1.10] | 0.875 | 1.11 [1.00–1.23] | 0.054 |
| Other | 0.75 [0.69–0.81] | <0.001 | 0.78 [0.71–0.85] | <0.001 |
| Unknown | 0.20 [0.08–0.53] | 0.001 | 0.26 [0.10–0.70] | 0.008 |
| Grade | | | | |
| I | 1 | Ref. | 1 | Ref. |
| II | 1.55 [1.33–1.82] | <0.001 | 1.28 [1.09–1.50] | 0.002 |
| III | 2.24 [1.93–2.61] | <0.001 | 1.51 [1.29–1.76] | <0.001 |
| IV | 2.18 [1.64–2.90] | <0.001 | 1.44 [1.08–1.92] | 0.013 |
| Unknown | 1.23 [0.98–1.55] | 0.068 | 1.04 [0.83–1.31] | 0.744 |
| Tumor size | | | | |
| <5 cm | 1 | Ref. | 1 | Ref. |
| ≥5 cm | 1.68 [1.57–1.80] | <0.001 | 1.10 [1.02–1.19] | 0.014 |
| Unknown | 1.14 [1.01–1.27] | 0.029 | 1.09 [0.97–1.22] | 0.154 |
| Primary site | | | | |
| Body | 1 | Ref. | 1 | Ref. |
| Antrum/pylorus | 1.14 [1.00–1.30] | 0.052 | 1.04 [0.91–1.19] | 0.525 |
| Cardia/fundus | 1.26 [1.11–1.44] | <0.001 | 1.39 [1.22–1.59] | <0.001 |
| Greater curvature | 1.21 [0.99–1.48] | 0.057 | 1.01 [0.83–1.24] | 0.920 |
| Lesser curvature | 1.10 [0.94–1.28] | 0.235 | 1.07 [0.92–1.25] | 0.382 |
| Overlapping regions | 1.39 [1.17–1.65] | <0.001 | 1.03 [0.87–1.23] | 0.742 |
| Stomach NOS | 1.31 [1.10–1.56] | 0.002 | 1.14 [0.96–1.36] | 0.141 |
| 8th AJCC T stage | | | | |
| T1 | 1 | Ref. | 1 | Ref. |
| T2 | 1.58 [1.39–1.79] | <0.001 | 1.26 [1.08–1.47] | 0.003 |

(Continued)

| Table 1 (continued) | | | | |
|---|---|---|---|---|
| (OS) Characteristics | Univariate analysis | | Multivariate analysis | |
| | HR (95% CI) | *p*-value | HR (95% CI) | *p*-value |
| T3 | 2.68 [2.43–2.95] | <0.001 | 1.58 [1.33–1.88] | <0.001 |
| T4a | 4.20 [3.75–4.71] | <0.001 | 2.01 [1.66–2.44] | <0.001 |
| T4b | 4.83 [4.14–5.63] | <0.001 | 2.41 [1.93–3.02] | <0.001 |
| 8th AJCC TNM stage | | | | |
| I | 1 | Ref. | 1 | Ref. |
| II | 1.93 [1.74–2.14] | <0.001 | 1.34 [1.14–1.58] | <0.001 |
| III | 3.87 [3.53–4.24] | <0.001 | 1.70 [1.40–2.07] | <0.001 |
| Chemotherapy | | | | |
| No/Unknown | 1 | Ref. | 1 | Ref. |
| Yes | 1.28 [1.20–1.37] | <0.001 | 0.76 [0.70–0.84] | <0.001 |
| Radiotherapy | | | | |
| No/Unknown | 1 | Ref. | 1 | Ref. |
| Yes | 1.28 [1.19–1.37] | <0.001 | 0.98 [0.90–1.06] | 0.576 |
| The number of ELNs | 0.99 [0.99–1.00] | <0.001 | 0.98 [0.98–0.99] | <0.001 |
| The number of pLNs | 1.06 [1.06–1.06] | <0.001 | 1.03 [1.02–1.04] | <0.001 |
| LNR | 7.84 [7.05–8.72] | <0.001 | 2.51 [2.04–3.08] | <0.001 |

distributed as 3,192 (50.26%) for N0, 1,204 (18.96%) for N1, 1,023 (16.11%) for N2, 328 (5.16%) for N3a, and 604 (9.51%) for N3b.

Patients were initially classified into AJCC stage categories of I, II, and III with percentages of 28.99%, 28.61%, and 42.40% respectively. The patients were then regrouped using rAJCC staging system resulting in 32.96% classified as rI, 35.18% as rII, and 31.87% as rIII. atients from SEER validation set, TCGA validation set, and our single center were regrouped subsequently according to the modified N classification.

The revised LNR-based rAJCC system, as illustrated in Fig. 2, it has validated the clinical relevance and effectiveness. The rAJCC system effectively stratifies patient prognoses across all evaluated datasets based on the LNR-based N classification (panels a, c, e, g), exhibiting statistically significant differences ($p < 0.001$). When patients were regrouped according to the rAJCC staging system, survival plots (panels b, d, f, h) remained distinct and showed significant statistical differences ($p < 0.001$). The detailed OS data and comparison results were shown in Table S3.

When evaluating the performance of two models, a lower AIC value suggests a model that fits the data well while balancing model complexity, whereas a lower BIC value implies a more stringent control over complexity, thus reducing the risk of overfitting. In our analysis, the rAJCC staging system demonstrated superior statistical performance over the 8th AJCC staging system, with both AIC (57,042.6 *vs*. 57,278.3) and BIC (57,054.9 *vs*. 57,290.7) values being lower in the primary training dataset. In the current study, we chose to depict the AUC across all time intervals using box plots, rather than focusing solely on the 3-year and 5-year AUCs. This methodological choice provides a more nuanced view of

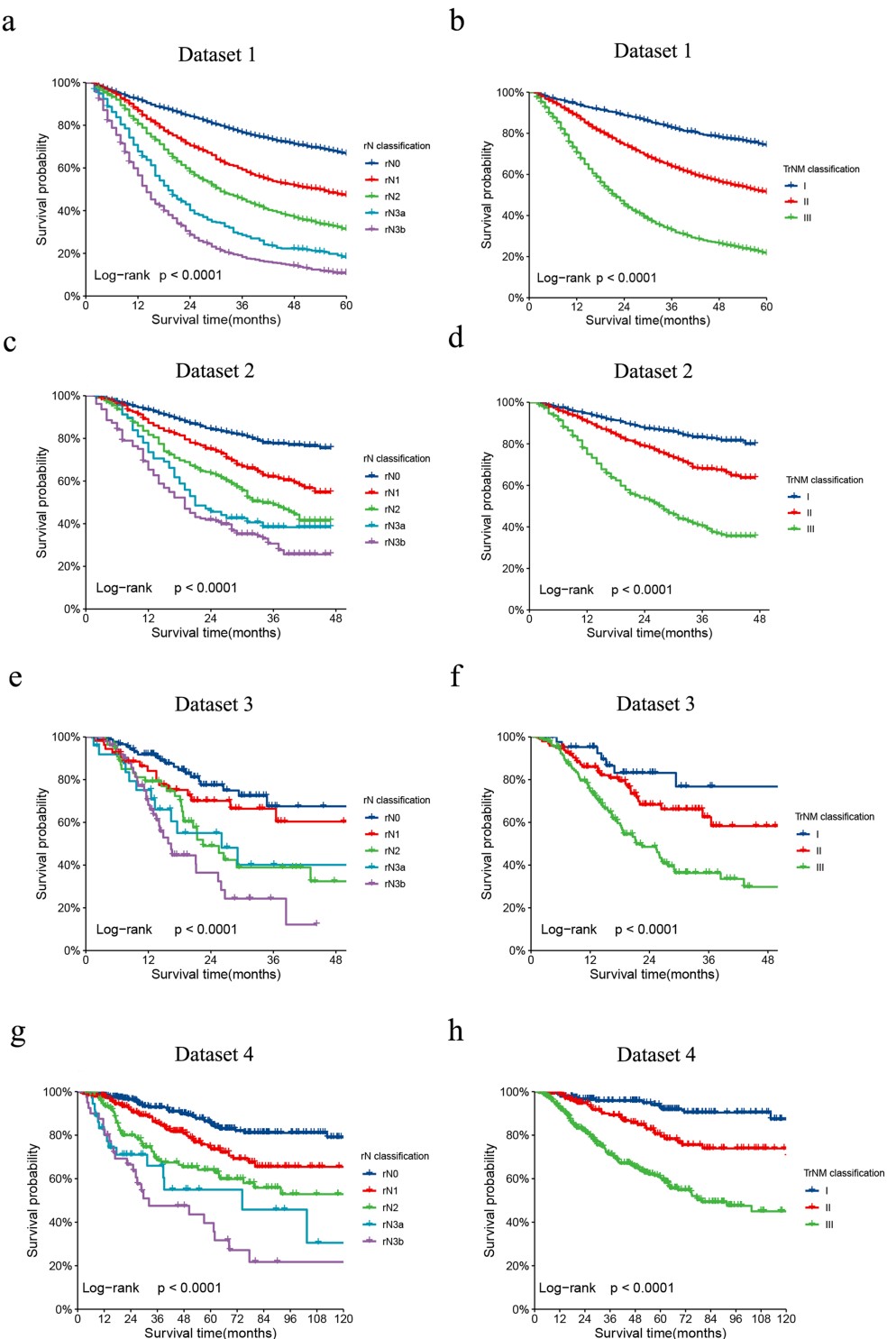

**Figure 2 Comparison of Kaplan-Meier survival curves for four datasets depicted according to the rN classification or rAJCC staging system.** The Kaplan-Meier survival curves of patients with gastric cancer in the (A) dataset 1, (C) dataset 2, (E) dataset 3 and (G) dataset 4 were depicted according to the rN classification. The Kaplan-Meier survival curves of patients with gastric cancer in the (B) dataset 1, (D) dataset 2, (F) dataset 3 and (H) dataset 4 were depicted according to the rAJCC staging system.

a

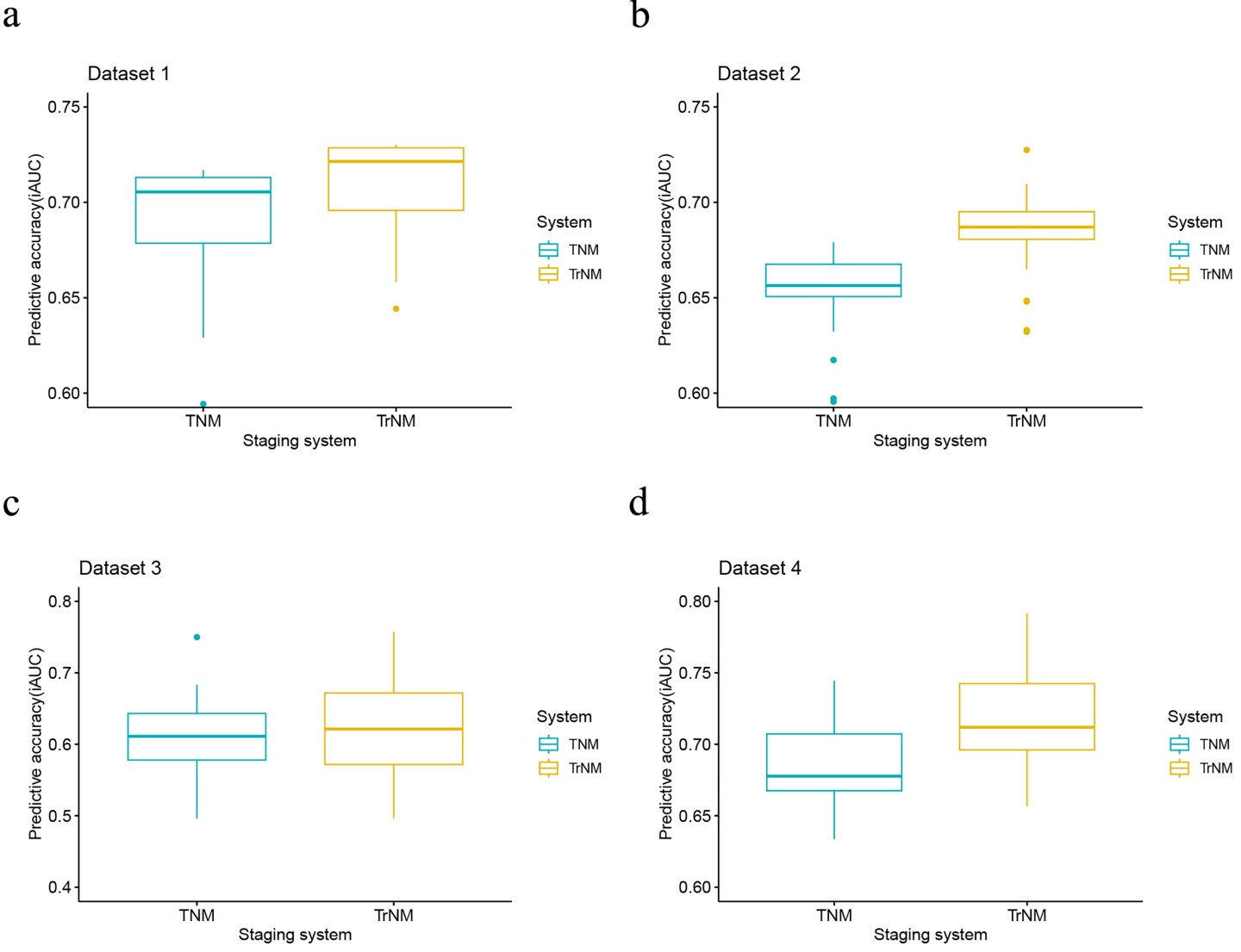

**Figure 3 Performance of the rAJCC staging systems compared with the 8th AJCC staging system in four datasets.** Performance of the rAJCC staging systems compared with the 8th AJCC staging system in the (A) dataset 1, (B) dataset 2, (C) dataset 3 and (D) dataset 4.

the rAJCC staging system's predictive capabilities over time, as opposed to the 8th edition AJCC staging system for gastric cancer, across multiple datasets. The rAJCC staging system showed a significantly higher AUC (73%, CI [71.7–74.2%]) compared to the 8th AJCC staging system (71.7%, CI [70.4–72.9%]) in primary training dataset 1 indicating greater discriminatory power, as illustrated in Fig. 2 and Table S4.

Further validation of the model using validation set from SEER (2016–2017), data from The Cancer Genome Atlas (TCGA), and a single-center dataset consistently demonstrated the rAJCC staging system's enhanced discriminatory power. This finding is corroborated by the visual representation in Fig. 3 and the numerical data presented in Table 2.

**Table 2 Comparison of the performance of the 8th AJCC and rAJCC classifications in the datasets.**

| Staging system | AIC | BIC |
|---|---|---|
| Dataset 1 | | |
| AJCC | 5,7278.3 | 57,290.7 |
| rAJCC | 5,7042.6 | 57,054.9 |
| Dataset 2 | | |
| AJCC | 8,291.7 | 8,300.4 |
| rAJCC | 8,222.2 | 8,230.9 |
| Dataset 3 | | |
| AJCC | 1,061.6 | 1,066.9 |
| rAJCC | 1,056.4 | 1,061.8 |
| Dataset 4 | | |
| AJCC | 1,966.4 | 1,972.6 |
| rAJCC | 1,947.5 | 1,953.7 |

## Ideal number of ELNs analysis based on LN involvement status and survival

The ELN count was assumed to be similar between radical total and distal gastric resection due to insufficient data on the surgical procedures, despite the theoretical difference in LN numbers harvested from the different lymphadenectomy ranges. The optimal threshold for ELN count analysis was established through a two-step approach. COX regression was initially applied, based on OS and DSS, to assess the HR for each ELN count in patients with negative LNs, using patients with a single ELN as the reference. The HR values for each ELN count group were analyzed with LOWESS scatter curve fitting. The Chow test identified the minimum ELN count where the curve's slope indicated a significant change. Logistic regression was then conducted on data from patients with negative or single LN metastasis, defining node status as the outcome variable and patients with a single ELN as the reference. The LOWESS and Chow test were utilized to plot the odds ratio curve for each ELN count group, identifying the ideal cutoff LN count for maximizing the detection efficacy of at least one involved LN.

Patients with negative LN was analyzed first, multivariate analysis have proved a greater number of LN was examined, the risk of potentially positive LN decreases thus improved the prognosis. HR of each ELN count compared with only one ELN (as reference) was analyzed using Cox proportional hazards regression model to determine the effect of ELN number on OS/DSS, after adjusting for other significant prognostic factors. LOWESS analysis was utilized to visualize the survival curves. Structural break ELN count was determined by Chow test as significant curvature change ($p < 0.05$) in the HR curve, which was minimum ELN count to obtain survival benefit, as shown in Figs. 4A, 4C. In the current study, curve of OS revealed 30 ELNs at least to guarantee survival benefit while which was 30 in DSS curve in Figs. 4B, 4D.

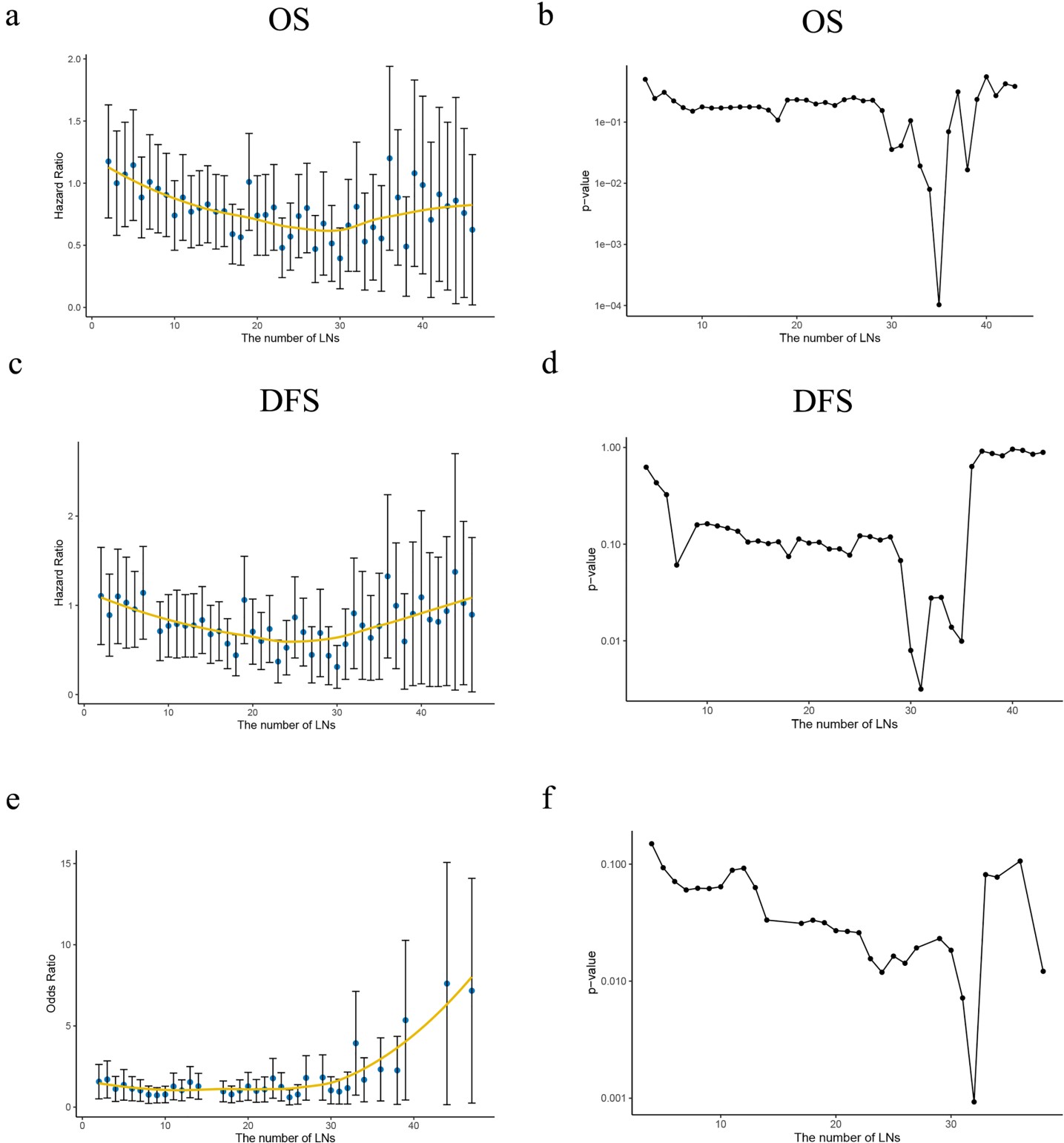

**Figure 4 Ideal number of ELNs analysis based on HR/OR values of each ELN count on OS/DSS.** (A, C) The HR values of the number of retrieved lymph nodes (LNs) from the Cox multivariate regression were fitted by the Lowess curves. (B, D) Describing the slope change of Lowess curves with Chow tests. (E) The OR values of the number of retrieved LNs from the Logistic multivariate regression were fitted by the Lowess curve. (F) Describing the slope change of Lowess curve 'e' with Chow tests.

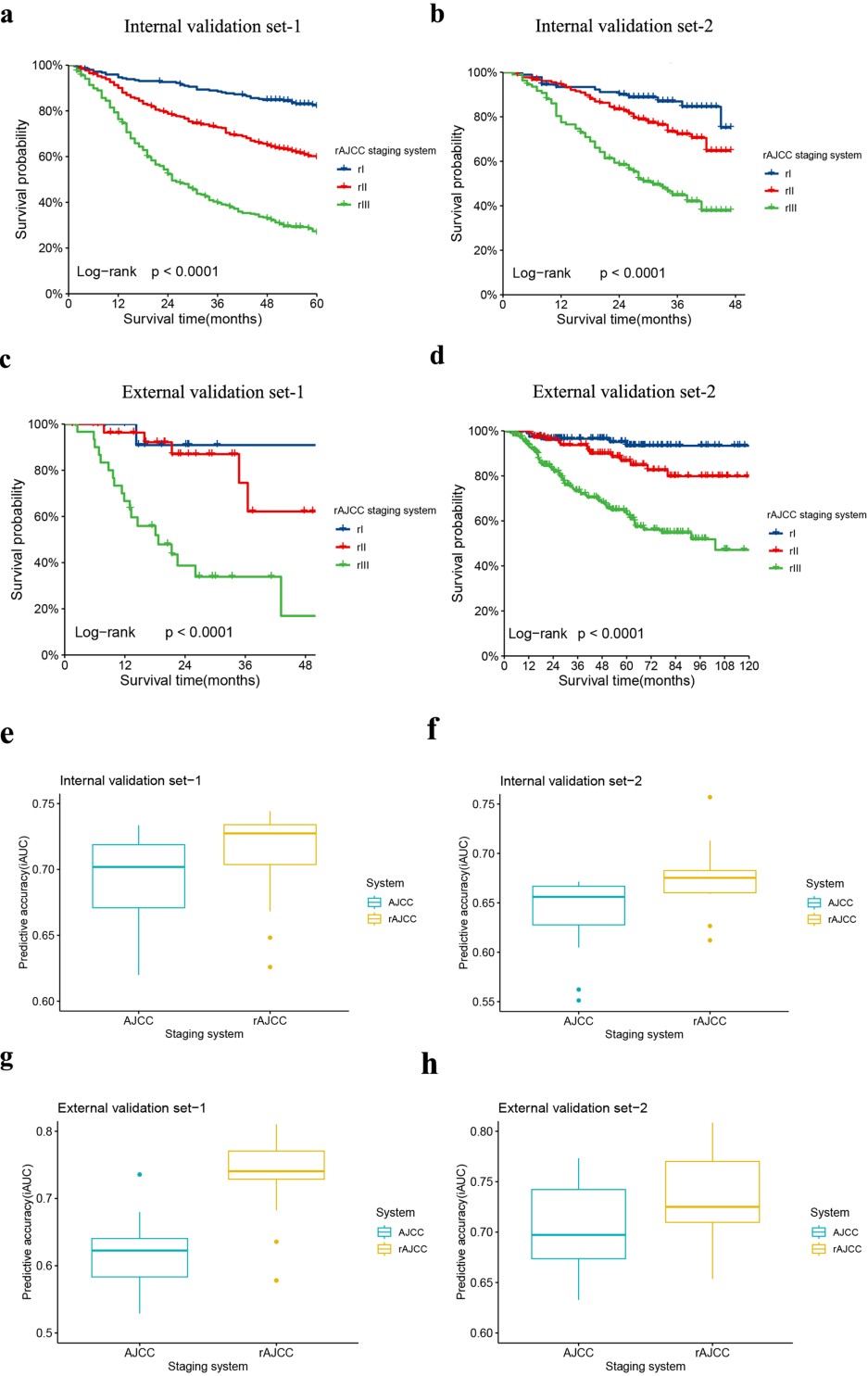

**Figure 5 Comparison of Kaplan-Meier survival curves for four validation sets depicted according to the rAJCC staging system.** The Kaplan-Meier survival curves of patients with gastric cancer in the (A) Internal validation set-1, (B) Internal validation set-2, (C) External validation set-1 and (D) External validation set-2 were depicted according to the rAJCC staging system.Performance of the rAJCC staging systems compared with the 8th AJCC staging system in the (E) Internal validation set-1, (F) Internal validation set-2, (G) External validation set-1 and (H) External validation set-2.

**Table 3 Comparison of the performance of the 8th AJCC and rAJCC staging systems in the validation sets in patient with ELN count ≥ 30.**

| Staging system | AIC | BIC |
|---|---|---|
| Internal validation set-1 | | |
| AJCC | 6,168.6 | 6,177.0 |
| rAJCC | 6,143.5 | 6,151.9 |
| Internal validation set-2 | | |
| AJCC | 1,270.2 | 1,275.7 |
| rAJCC | 1,257.5 | 1,263.0 |
| External validation set-1 | | |
| AJCC | 185.5 | 188.1 |
| rAJCC | 172.6 | 175.2 |
| External validation set-2 | | |
| AJCC | 978.6 | 983.6 |
| rAJCC | 966.0 | 971.1 |

With regard to the correlation between ELN count and positive LNs identified, a greater ELN count was associated with a greater number of PLNs. Patients from SEER (2010–2015) cohort was then redistributed into node-negative and only one positive LN group. LN status was then assessed by correlating the ELN number and the proportion of each node stage category (node negative *vs*. one node positive) by using a binary logistic regression model after adjusting for other potential confounders. The curves of odds ratios (ORs; LN involvement) were fitted by using a LOWESS smoother. The minimum ELN count in detecting at least one involved LN was clarified using Chow test *(p < 0.05)* as 29 lymph nodes. Both approaches indicated a minimum ELN of 30 was necessary. All curved and Chow test results were shown in Fig. 4.

## Integration and validation of both ideal ELN count and LNR GC staging system

Patients with an ELN count of 30 or more were identified from each dataset, were categorized as Internal validation set-1, Internal validation set-2, External validation set-1, and External validation set-2, and subsequently reassessed using the rAJCC staging system. All validation datasets, both internal and external, showed a significantly different prognosis *(p < 0.001)* and improved HR performance compared to the 8th edition AJCC system in patients with ELN > 30, as illustrated in Fig. 5 (panels a, b, c, d) and detailed in Table S5. The AUC of the novel classification significantly exceeded that of the 8th AJCC classification, with a clear advantage in terms of AIC and BIC, as depicted in Fig. 5 (panels e, f, g, h), and summarized in Table 3. Furthermore, for patients with ELN < 30, the rAJCC staging system still showed superiority than the 8[th] AJCC system, as shown in Fig. S1.

Furthermore, the integrated model, which combines the ELN count with the rAJCC staging system, was compared with the original rAJCC model without ELN count integration. Given the inability to evaluate long-term survival rates for patients registered between 2016 and 2017 in the SEER database, the comparative analysis was restricted to

the TCGA database and our single-center dataset. The findings indicated that the integrated model achieved a higher AUC, outperforming the rAJCC system in terms of predictive accuracy, as demonstrated in Fig. S2.

## DISCUSSION

The current study found that LNR and ELN count are both associated with the prognosis of gastric cancer patients, consistent with previous studies. Higher ELN count suggests less residual micro-metastatic disease and can aid in precise staging, leading to more effective adjuvant therapy and improved outcomes. LNR was also shown to be a more accurate prognostic factor than AJCC N classification. LNR could be easily flattered when ELN was inadequate, but previous studies did not analyze ELN count as a parameter in revised staging system construction. An integration GC staging system for nonmetastatic GC was developed using LNR and ELN count incorporated into the 8th AJCC staging system. Results indicate that the integration GC staging system provides better prognosis and discriminatory capacity compared to the 8th AJCC staging system, as evidenced by improved AUC square and better AIC/BIC value.

LNR was analyzed variously in previous studies. The classification applied by *Zeng et al. (2023a)*, and *He et al. (2022)* divided lymph node status into three categories: low LNR (0%–20%/25%), middle-LNR (20/25%–50%), high-LNR (>50%). *Kano et al. (2020)* and *Jiang et al. (2022)* classifications stratified patients into three subgroups with different cut-offs: N0 (0%–10/30%), N1 (10/30%–25/45%), N3 (>30/45%). The classification applied by *Zhang et al. (2014)* and *Chen et al. (2022)* stratified patients into four groups: N0 (0%), N1 (1–20%), N2 (21–50%/69%) and N3 (>50%/70%). In this study, we initially conducted statistical analyses using data from dataset 1 to precisely determine the threshold values for the LNR to define a novel N stage. Subsequently, the 8th edition of the AJCC N classification was replaced with the rN classification, leading to the development of an enhanced rAJCC staging system. Patients were then reclassified using this rAJCC staging system. Survival data analysis demonstrated that all pairwise comparisons revealed significant differences in prognosis for patients with the modified rN classification and the rAJCC staging system. The system's discriminatory power was meticulously assessed through the application of the AIC, BIC and AUC. The AIC and the BIC are recognized as essential tools in the statistical analysis of regression models, including both linear and logistic regression. These criteria play a pivotal role in the model selection process, providing a quantitative framework for comparing the fit of different models. While the AUC is a relevant metric for evaluating the performance of classification models. In addition, our analysis deviated from conventional methods by utilizing a box plot representation of the AUC across all time intervals derived from the rAJCC staging system. This approach differs from the traditional comparison of solely the 3-year and 5-year AUCs, offering a more comprehensive perspective on the system's predictive capabilities over time. The median AUC was chosen as the evaluative metric, providing a cumulative measure of predictive performance across all considered time points. By using the median, our analysis is less affected by extreme values or outliers that could skew the results if only specific time intervals were examined. This approach allows for a more refined assessment

of the model's predictive power, reflecting its performance throughout the entire follow-up period. In the context of model comparison using AIC and BIC, the rAJCC model exhibited lower values for both, signifying its superiority. The median AUC, as an aggregate measure of predictive performance, also displayed a higher value, indicating enhanced discrimination capability across all datasets. However, relying solely on LNR to determine pathological staging may be inaccurate due to incomplete lymphadenectomy, requiring a combined assessment of lymph node dissection.

ELN count is an independent prognostic factor in multiple cancers, including GC, and higher ELN counts are associated with more accurate nodal staging and improved survival. The count of ELNs can significantly impact the survival outcomes of patients with gastric cancer (GC). This is because it directly affects the accuracy of staging, allowing for more personalized postoperative treatment and ultimately leading to improved survival rates. Additionally, ELN count can serve as a valuable indicator of surgical quality, as a higher count reduces the likelihood of residual positive lymph nodes (PLNs) and nodal micro-metastases. Ultimately, this lowers the risk of postoperative recurrence. Various factors can affect the number of ELNs in gastrectomy, such as the surgical technique, extent of surgery, diligence and thoroughness of pathologic examination, condition of specimens, and innate number of LNs for each patient. While AJCC recommends a minimum of 16 ELNs for accurate staging, there is no consensus on the optimal threshold number of ELNs to address both stage migration and long-term survival.

Research has suggested that patients with a higher number of examined lymph nodes may have better survival due to stage migration and more accurate selection for adjuvant systemic therapy. *Zhao et al. (2023)*. recommended ≥24 ELNs in patients with advanced GC and demonstrated that the AJCC recommendation of ≥16 ELNs was insufficient for determining the N stage. *Guo et al. (2021)*. further pointed out that ≥27 ELNs was associated with a maximum survival advantage in patients with GC undergoing surgery, using the SEER database and 144 patients from China. *Zhang et al. (2020)* and *Huang et al. (2021)* reported a minimum of 31 and 33 ELN counts would improve the prognosis according to data from SEER database and institutions from China. The current study aimed to determine the optimal cutoff value of ELN by considering two requirements: ensuring maximum improvement in the prognosis of node-negative patients without downgrading due to potential metastasis and maximizing the detection efficiency of at least one involved LN. In this study, COX regression was used to examine the prognostic value of ELN count in patients with negative LNs. Patients with only one ELN were used as a reference, and the HR value for each ELN count group for both OS and DSS was analyzed using LOWESS scatter curve fitting. The minimum count of ELNs was determined by Chow test at which the slope of the curve changes significantly. Logistic regression analysis was performed on patients with negative lymph nodes and only one LN metastasis from the SEER database, using node status as the outcome variable. The reference value was set as patients with only one ELN. LOWESS and chow test help draw the curve and defined as the ideal cutoff LN count for maximum detection efficacy of at least one involved LN. Both approaches suggested that 30 or more ELNs contributed to maximum improvement in the prognosis of node-negative patients and earlier detection of at least one LN metastasis.

The integration of revised staging system incorporating LNR and ELN count was conducted and evaluated. Patients with more than 30 ELNs were included and regrouped according to the cut-offs of LNR as mentioned above. The integrated system presents valuable criteria for identifying patients with a favorable prognosis, thereby preventing overtreatment. Conversely, patients with poorer survival prospects can be treated with a more intensive approach. Additionally, the integrated staging scheme is likely to be clinically feasible since the enhancement in prognostic accuracy does not result in increased complexity. For patients with ELN < 30, the current rAJCC system can provide pathological staging for risk and prognosis assessment, but additional analysis is needed to determine the risk of inadequate lymphadenectomy and the need for adjuvant therapy.

The log odds of positive lymph nodes (LODDS) is a new method for assessing lymph node status in cancer (*Que et al., 2023*). Studies have shown that LNR and LODDS perform similarly in predicting prognosis for gastric cancer patients when an adequate number of lymph nodes are harvested (*Cao et al., 2019*), but LODDS can be influenced by the total number of retrieved nodes (*Lai, Zheng & Li, 2022*; *Díaz Del Arco et al., 2024*). Although LODDS may have better predictive value as a continuous variable for disease-specific survival, it is difficult to interpret and use in clinical practice (*Lu et al., 2017*; *Wang et al., 2023*). Therefore, LNR was used as the parameter in the current analysis.

The current study has several limitations. Despite our diligent efforts to guarantee the accuracy and quality of data retrieved from the SEER database, there still exist concerns regarding data miscoding and insufficiency. In addition, surgical procedures, surgical instruments, surgical skills, examinations of lymph nodes and adjuvant chemotherapies changed during the evolution of the cohort, which may have influenced patients' prognosis; Furthermore, the incidence and treatment protocol for locally advanced GC of the same TNM category differs between Asian and Western, which may explain the lower 5-year OS rate in the SEER cohort compared with the Chinese cohort.

## CONCLUSION

The development of a novel GC staging system, which integrated the LNR-based N classification and the minimum ELN count, has exhibited superior prognostic accuracy, holding promise as a valuable asset in the clinical management of GC.

However, it is important to acknowledge that the accuracy and quality of data derived from large databases may introduce limitations. Despite these challenges, the integration of molecular biomarkers and genetic profiling, alongside personalized therapeutics and a global data collection strategy, is expected to yield a more accurate and effective staging system in the future.

### Funding

This study is supported by the Medical Scientific Research Foundation of Guangdong Province, China (No. A2021163) and the Clinical Medical Research 5010 Program of Sun

Yat-sen University (2018023). The funders had no role in study design, data collection and analysis, decision to publish, or preparation of the manuscript.

### Grant Disclosures
The following grant information was disclosed by the authors:
Medical Scientific Research Foundation of Guangdong Province, China: A2021163.
Clinical Medical Research 5010 Program of Sun Yat-sen University: 2018023.

### Competing Interests
The authors declare that they have no competing interests.

### Author Contributions
- Guiru Jia conceived and designed the experiments, performed the experiments, analyzed the data, prepared figures and/or tables, authored or reviewed drafts of the article, and approved the final draft.
- Dagui Zhou performed the experiments, prepared figures and/or tables, collect data, and approved the final draft.
- Xiao Tang performed the experiments, prepared figures and/or tables, collect data, and approved the final draft.
- Jianpei Liu performed the experiments, authored or reviewed drafts of the article, collect data, and approved the final draft.
- Purun Lei conceived and designed the experiments, performed the experiments, analyzed the data, prepared figures and/or tables, and approved the final draft.

### Human Ethics
The following information was supplied relating to ethical approvals (*i.e.*, approving body and any reference numbers):

The protocol for this research project has been approved by Ethics Committee of The Third Affiliated Hospital of Sun-Yat sen University.

### Data Availability
The data is available at Zenodo: Purun, L. (2024). Prognostic value of a modified pathological staging system for gastric cancer based on the number of retrieved lymph nodes and metastatic lymph node ratio raw data [Data set]. Zenodo. https://doi.org/10.5281/zenodo.13335334.

### Supplemental Information
Supplemental information for this article can be found online at http://dx.doi.org/10.7717/peerj.18165#supplemental-information.

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
