# Peer review of "Prognostic value of a modified pathological staging system for gastric cancer based on the number of retrieved lymph nodes and metastatic lymph node ratio"

_PeerJ, doi:10.7717/peerj.18165_

## Round 0.1 · original submission · Major Revisions

I kindly request that the article be seriously revised according to the opinions of the reviewers.

·

Basic reporting

No comment

Experimental design

No comment

Validity of the findings

No comment

Additional comments

The article presents a new gastric cancer staging method that considers the lymph node ratio (LNR) and the number of examined lymph nodes (ELN). The research found a strong link between LNR, ELN count, and patient outcomes. A revised AJCC staging system, the rAJCC, was introduced, which showed better predictive accuracy than the current system. The study highlights the significance of incorporating LNR and ELN count in gastric cancer treatment and management.
There are some problems, which must be solved before it is considered for publication. If the following problems are well-addressed, we believes that the essential contribution of this paper are important for gastric cancer treatment.
1.The study did not provide detailed descriptions of the surgical procedures and specific details of adjuvant treatments, which may affect the interpretation of the results.
2.Additionally, it is important for the study to consider the variability among patients of different ethnicities, regions, and economic statuses for its global applicability.
3.The study found that the minimum number of Examined Lymph Nodes (ELN) is 30, which aligns with the requirements proposed in some current guidelines. The article only analyzes the effectiveness of the new system in cases where ELN is greater than 30 and does not address cases with less than 30 ELN. It is recommended to include this section to make the article more comprehensive.
4.In the article and supplementary materials, only the survival curves for the rN and 8th edition AJCC TNM systems across different datasets are provided. It is suggested to include survival curves for the 8th edition AJCC N staging and TNM systems for comparison to demonstrate the superiority of the rN system over the N staging system.
5.In the supplementary document, Supplementary Table 4 appears to be the same as Table 3. Please correct and resubmit.
6.In the DISCUSSION section, line 253, the statement "The rAJCC system replaces LNR classification with AJCC N classification and requires a minimal ELN of over 30 for adequate lymphadenectomy" should be checked for a potential error. It should perhaps read, "The rAJCC system replaces AJCC N classification with LNR classification..."
7.Another obvious problem with this paper is lack of sufficient explanation of the results. You need to explain your results in detail.
8.CONCLUSIONS needs more in it, the authors are suggested to highlight important findings and include afterthought of this work.
9.The references cited in the study are mostly from publications 5-10 years ago and do not represent the latest developments in the field. It is recommended to include studies related to gastric cancer ELN and LNR from within the last five years to substantiate the innovation and value of this research”.

Reviewer 2 ·

Basic reporting

From the text: Gastric cancer is the sixth most common cancer globally and the second most common malignancy in China.
--Please provide up-to-date references.
Bray F, Laversanne M, Sung H, Ferlay J, Siegel RL, Soerjomataram I, Jemal A. Global cancer statistics 2022: GLOBOCAN estimates of incidence and mortality worldwide for 36 cancers in 185 countries. CA Cancer J Clin. 2024 May-Jun;74(3):229-263. doi: 10.3322/caac.21834. Epub 2024 Apr 4. PMID: 38572751.

Experimental design

From the text: In the current study, the SEER database was used for establishing a staging system by replacing the 8th AJCC N classification with the LNR classification and incorporating minimum ELN count.
From the text: Clinical data from the US Surveillance, Epidemiology, and End Results (SEER) Program from 2010-2015 (https://seer.cancer.gov/) was extracted and analyzed as training set, data from 2016-2017 was adopted as internal validation set.
--The authors cited AJCC 2017 (Eighth Edition AJCC Cancer Staging Guidelines), but the patients were selected from the period prior to the publication of these guidelines.
(Edge, S.B., Compton, C.C. The American Joint Committee on Cancer: the 7th Edition of the AJCC Cancer Staging Manual and the Future of TNM. Ann Surg Oncol 17, 1471–1474 (2010). https://doi.org/10.1245/s10434-010-0985-4)

Validity of the findings

--There are no references from the last two years among your references. In the discussion section please also discuss the information of the current references.

Additional comments

--I would like to congratulate the authors on their works.
--I have the following comments/concerns regarding the study, which authors may find useful.

From the text: Abstract: Given the potential benefits of using LNR for improved prognostication and treatment planning.
--The first time you use an abbreviation in the text and tables, present both the spelled-out version and the short form.

From the text: Gastric cancer is the sixth most common cancer globally and the second most common malignancy in China.
--Please provide up-to-date references.
Bray F, Laversanne M, Sung H, Ferlay J, Siegel RL, Soerjomataram I, Jemal A. Global cancer statistics 2022: GLOBOCAN estimates of incidence and mortality worldwide for 36 cancers in 185 countries. CA Cancer J Clin. 2024 May-Jun;74(3):229-263. doi: 10.3322/caac.21834. Epub 2024 Apr 4. PMID: 38572751.

From the text: In the current study, the SEER database was used for establishing a staging system by replacing the 8th AJCC N classification with the LNR classification and incorporating minimum ELN count.
From the text: Clinical data from the US Surveillance, Epidemiology, and End Results (SEER) Program from 2010-2015 (https://seer.cancer.gov/) was extracted and analyzed as training set, data from 2016-2017 was adopted as internal validation set.
--The authors cited AJCC 2017 (Eighth Edition AJCC Cancer Staging Guidelines), but the patients were selected from the period prior to the publication of these guidelines.
(Edge, S.B., Compton, C.C. The American Joint Committee on Cancer: the 7th Edition of the AJCC Cancer Staging Manual and the Future of TNM. Ann Surg Oncol 17, 1471–1474 (2010). https://doi.org/10.1245/s10434-010-0985-4)

--There are no references from the last two years among your references. In the discussion section please also discuss the information of the current references.
-Ergenç, M., Uprak, T.K., Akın, M.İ. et al. Prognostic significance of metastatic lymph node ratio in gastric cancer: a Western-center analysis. BMC Surg 23, 220 (2023). https://doi.org/10.1186/s12893-023-02127-y (reviewer's own article)
--Please note that citations recommended by reviewer may be included if you believe that they add value to your manuscript. If you do not believe that such citations would benefit your manuscript, then please provide explanation(s) in your response letter.
-Yin K, Jin X, Pan Y, Zi M, Zheng Y, Ma Y, Pang C, Liu K, Chen J, Wei Y, Liu D, Cheng X, Yuan L. Revolutionizing T3-4N0-2M0 gastric cancer staging with an innovative pathologic N classification system. J Gastrointest Surg. 2024 May 29:S1091-255X(24)00486-4. doi: 10.1016/j.gassur.2024.05.031. Epub ahead of print. PMID: 38821213.

From the text: The rAJCC staging system exhibited a significantly higher level of discriminatory power compared to the 8th AJCC staging system, as evident in the higher AUC value (73.0, CI [71.7,74.2] vs 71.7 CI [70.4,72.9]) and outperformed the 8th AJCC staging system with respect to the AIC (57042.6 vs. 57278.3) and BIC (57054.9 vs 57290.7).
--“AUC is an effective way to summarize the overall diagnostic accuracy of the test. It takes values from 0 to 1, where a value of 0 indicates a perfectly inaccurate test and a value of 1 reflects a perfectly accurate test. AUC can be computed using the trapezoidal rule. In general, an AUC of 0.5 suggests no discrimination (i.e., ability to diagnose patients with and without the disease or condition based on the test), 0.7 to 0.8 is considered acceptable, 0.8 to 0.9 is considered excellent, and more than 0.9 is considered outstanding.”
-Mandrekar JN. Receiver operating characteristic curve in diagnostic test assessment. J Thorac Oncol. 2010 Sep;5(9):1315-6. doi: 10.1097/JTO.0b013e3181ec173d. PMID: 20736804.
--Please specify the AUC value as a percentage.

---

## Round 0.2 · accepted · Accept

Following the incorporation of the recommended revisions, I believe the article is now suitable for publication.

Reviewer 2 ·

Basic reporting

I want to thank the authors for their responses and revisions. After these revisions and additions, I think the article has improved and is suitable for publication.

Experimental design

I want to thank the authors for their responses and revisions. After these revisions and additions, I think the article has improved and is suitable for publication.

Validity of the findings

I want to thank the authors for their responses and revisions. After these revisions and additions, I think the article has improved and is suitable for publication.

Additional comments

I want to thank the authors for their responses and revisions. After these revisions and additions, I think the article has improved and is suitable for publication.